# Mevalonate Pathway Enzyme HMGCS1 Contributes to Gastric Cancer Progression

**DOI:** 10.3390/cancers12051088

**Published:** 2020-04-27

**Authors:** I-Han Wang, Tzu-Ting Huang, Ji-Lin Chen, Li-Wei Chu, Yueh-Hsin Ping, Kai-Wen Hsu, Kuo-Hung Huang, Wen-Liang Fang, Hsin-Chen Lee, Chian-Feng Chen, Chen-Chung Liao, Rong-Hong Hsieh, Tien-Shun Yeh

**Affiliations:** 1Program in Molecular Medicine, National Yang-Ming University and Academia Sinica, Taipei 112, Taiwan; ellennugget@gmail.com; 2Institute of Anatomy and Cell Biology, School of Medicine, National Yang-Ming University, Taipei 112, Taiwan; tzu-ting.huang@nih.gov (T.-T.H.); Jlchen_@outlook.com (J.-L.C.); 3Department and Institute of Pharmacology, School of Medicine, National Yang-Ming University, Taipei 112, Taiwan; nealchu125@gmail.com (L.-W.C.); yhping@ym.edu.tw (Y.-H.P.); hclee2@ym.edu.tw (H.-C.L.); 4Research Center for Tumor Medical Science, China Medical University, Taichung 404, Taiwan; kwhsu@mail.cmu.edu.tw; 5Graduate Institutes of New Drug Development, China Medical University, Taichung 404, Taiwan; 6Institute of Clinical Medicine, School of Medicine, National Yang-Ming University, Taipei 112, Taiwan; khhuang@vghtpe.gov.tw (K.-H.H.); wlfang@vghtpe.gov.tw (W.-L.F.); 7Department of Surgery, Taipei Veterans General Hospital, Taipei 112, Taiwan; 8Cancer Progression Research Center, National Yang-Ming University, Taipei 112, Taiwan; cfchen@ym.edu.tw; 9Proteomics Research Center, National Yang-Ming University, Taipei 112, Taiwan; ccliao@ym.edu.tw; 10School of Nutrition and Health Sciences, College of Nutrition, Taipei Medical University, Taipei 110, Taiwan; hsiehrh@tmu.edu.tw; 11Institute of Biochemistry and Molecular Biology, National Yang-Ming University, Taipei 112, Taiwan; 12Graduate Institute of Medical Sciences, College of Medicine, Taipei Medical University, Taipei 110, Taiwan

**Keywords:** HMGCS1, mevalonate pathway, gastric cancer progression, integrated stress response, pluripotency

## Abstract

The 3-hydroxy-3-methylglutaryl-CoA synthase 1 (HMGCS1) is a potential regulatory node in the mevalonate pathway that is frequently dysregulated in tumors. This study found that HMGCS1 expression is upregulated in stomach adenocarcinoma samples of patients and tumorspheres of gastric cancer cells. HMGCS1 elevates the expression levels of the pluripotency genes Oct4 and SOX-2 and contributes to tumorsphere formation ability in gastric cancer cells. HMGCS1 also promotes in vitro cell growth and progression and the in vivo tumor growth and lung metastasis of gastric cancer cells. After blocking the mevalonate pathway by statin and dipyridamole, HMGCS1 exerts nonmetabolic functions in enhancing gastric cancer progression. Furthermore, the level and nuclear translocation of HMGCS1 in gastric cancer cells are induced by serum deprivation. HMGCS1 binds to and activates Oct4 and SOX-2 promoters. HMGCS1 also enhances the integrated stress response (ISR) and interacts with the endoplasmic reticulum (ER) stress transducer protein kinase RNA-like endoplasmic reticulum kinase (PERK). Our results reveal that HMGCS1 contributes to gastric cancer progression in both metabolic and nonmetabolic manners.

## 1. Introduction 

Gastric carcinoma, one of the most common cancers, ranks as the third most frequent cause of global cancer mortality [1]. The mechanisms responsible for the aggressiveness of gastric cancer have not yet been clearly characterized. The mevalonate pathway for cholesterol synthesis and protein prenylation is frequently dysregulated in tumors and has been implicated in cancer development and progression [2,3]. Mevalonate metabolism is also involved in the maintenance of basal breast cancer stem cells and is one of the potential therapeutic targets [4]. 3-hydroxy-3-methylglutaryl-coenzyme A (HMG-CoA) reductase (HMGCR), which converts HMG-CoA into mevalonate, is the rate-limiting enzyme of the mevalonate pathway [5]. Mounting evidence reveals that other enzymes besides HMGCR can also serve as controlling points of cholesterol synthesis [6]. HMG-CoA synthase 1 (HMGCS1) is the cytoplasmic enzyme immediately upstream of HMGCR in the mevalonate pathway and condenses acetyl-CoA and acetoacetyl-CoA to HMG-CoA [5]. It has been shown that HMGCS1 is one of the potential regulatory nodes in the mevalonate pathway [7].

Statins, U.S. Food and Drug Administration (FDA)-approved drugs, suppress HMGCR activity and are associated with reduced cancer-related mortality, including in gastric cancer patients [8,9]. However, some contradictory results found by meta-analyses for several cancers do not support the protective effect of statins [10]. Mevalonate pathway genes, including HMGCR and HMGCS1 in statin-insensitive multiple myeloma cells, are upregulated by statin treatment [11]. Both HMGCR and HMGCS1 levels are induced by sterol regulatory element-binding protein 2 (SREBP2) [6] but reduced by acyl-CoA binding protein (ACBP) [12]. Dipyridamole treatment suppresses the HMGCS1 level through inhibition of cleavage of SREBF2 [13] and downregulates the expression of HMGCS1, c-Myc, and cyclin D1 in colon cancer cells [14]. HMGCS1 expression is also controlled by mutant p53 [15]. Ferrous and ferric irons upregulate HMGCS1 expression in human vascular smooth muscle cells [16]. Mounting evidence shows the linkage of endoplasmic reticulum (ER) stress and lipid metabolism in diseases [17,18]. Homocysteine-induced ER stress activates SREBPs and induces the biosynthesis of cholesterol in cultured hepatocytes and smooth muscle cells [17]. STAT6 downregulation enhances cholesterol synthesis and ER stress-mediated apoptosis in lung cancer cells [18].

Analyzing cancer genomics datasets showed that the HMGCS1 gene is amplified in various cancers such as stomach, breast, and lung cancers [19]. High levels of HMGCS1 mRNAs are correlated with poor prognosis and lower survival rates in breast cancer patients [3]. Recent data from a meta-analysis suggested that HMGCS1 is one of the candidates involved in forming tumor stem-like breast cancer cells [20]. Knockdown of HMGCS1 significantly sensitized A549 cells to statin-mediated apoptosis [19]. Therefore, we investigated the biological functions of HMGCS1 in controlling tumor development and the progression of gastric cancer cells and delineated the underlying mechanisms of HMGCS1-mediated gastric cancer progression.

## 2. Results

### 2.1. HMGCS1 Expression is Upregulated in Stomach Adenocarcinoma Samples of Patients and Tumorspheres of Gastric Cancer Cells

HMGCS1, one of the potential regulatory nodes in the mevalonate pathway, has been implicated in controlling progression [2,3] and stemness [4] of cancer cells. Since the HMGCS1 gene is amplified in stomach cancer [19], we investigated whether HMGCS1 affects gastric cancer progression. To examine the clinical relevance of HMGCS1 mRNA expression, quantitative real-time PCR was first performed in gastric cancer samples and the corresponding adjacent normal tissues of gastric cancer patients. Levels of HMGCS1 mRNA in gastric tumor samples from patients were higher than those of adjacent nontumor samples (Figure 1A). Notably, HMGCS1 mRNA expression was dramatically upregulated in lymph node tumor samples from 26 gastric cancer patients with lymph node metastasis. According to stage classification, the mRNA expression of HMGCS1 was increased in stomach adenocarcinoma specimens from patients with advanced gastric cancer (stages I to III), compared with those of the adjacent nontumor tissues (Appendix A). The results of the Kaplan‒Meier survival plot showed that gastric cancer patients with higher levels of HMGCS1 mRNA had a poorer overall survival rate (Figure 1B).

Because more than 95% of tumors of stomach are adenocarcinomas, cell lines of human stomach adenocarcinoma were also examined. The results of Western blot analysis showed that HMGCS1 protein was differentially expressed in gastric cancer cells, including AGS, NUGC-3, KATO III, SNU-16, and NCI-N87 cells (Figure 1C). To check whether HMGCS1 is involved in regulating the stem cell-like phenotype, HMGCS1 expression in tumorspheres of gastric cancer cells was examined. Levels of mRNA (Figure 1D) and protein (Figure 1E) of HMGCS1 were enhanced in tumorspheres of KATO III and NCI-N87 gastric cancer cells compared with those in their parental cells according to quantitative real-time PCR and Western blot analysis, respectively.

### 2.2. HMGCS1 Elevates Levels of Pluripotency Genes Oct4 and SRY (Sex Determining Region Y)-Box 2 (SOX-2) and Contributes to Progression in Gastric Cancer Cells

To further investigate the roles of HMGCS1 in the progression of gastric cancer cells, overexpression of exogenous HMGCS1 and knockdown of endogenous HMGCS1 were induced in the present study. Therefore, we performed experiments using AGS, KATO III, and NCI-N87 cells moderately expressing the HMGCS1 protein level. The results showed that mRNA levels of pluripotency genes Oct4 and SOX-2 in AGS and NCI-N87 cells were promoted after transfecting HMGCS1-expressing plasmid construct (Figure 2A). The exogenous HMGCS1 also elevated protein levels of Oct4 and SOX-2 in AGS and KATO III cells, as shown by Western blot analysis (Figure 2B). Tumorsphere formation in KATO III and NCI-N87 cells also increased after transfecting the HMGCS1-expressing construct (Figure 2C).

Consistently, the data showed that mRNA levels of Oct4 and SOX-2 in NCI-N87 cells were suppressed after infection with lentiviruses expressing small interfering RNAs (siRNAs) against HMGCS1 (Appendix A). HMGCS1 knockdown decreased the protein levels of Oct4 and SOX-2 in AGS and NCI-N87 cells (Appendix A). Tumorsphere formation in NCI-N87 cells was reduced after infecting lentiviruses expressing siRNAs against HMGCS1 (Appendix A).

Next, the effect of HMGCS1 on the growth and progression of gastric cancer cells was analyzed. The cumulative cell numbers of KATO III cells transfected with the HMGCS1-expressing construct were elevated compared with those of control cells by the trypan blue exclusion method (Figure 2D). On the contrary, the cumulative cell numbers of NCI-N87 cells were diminished after infecting lentiviruses expressing siRNAs against HMGCS1 (Appendix A). The results of 3-(4,5-dimethyl-2-thiazolyl)-2,5-diphenyl tetrazolium bromide (MTT) assay showed that the growth of AGS, KATO III, and NCI-N87 cells was enhanced by transfecting the HMGCS1-expressing construct for 48 h (Figure 2E); in contrast, the growth of AGS and NCI-N87 cells was inhibited by infecting lentiviruses expressing siRNAs against HMGCS1 (Appendix A). Overexpression of HMGCS1 increased colony formation in KATO III and NCI-N87 cells in a colony formation assay (Figure 2F), but knockdown of HMGCS1 decreased this ability in NCI-N87 cells (Appendix A). Both the cell migration and invasion of AGS, KATO III, and NCI-N87 cells were promoted by HMGCS1 overexpression (Figure 2G). As expected, both abilities were suppressed by HMGCS1 knockdown in AGS and NCI-N87 cells (Appendix A).

### 2.3. HMGCS1 Enhances Tumor Growth and Lung Metastasis of Gastric Cancer Cells

Mouse models were employed to further evaluate the effect of HMGCS1 on tumor growth and the metastatic colonization of gastric cancer cells. After subcutaneously injecting NCI-N87 cells into nude mice, the xenografted tumor sizes of NCI-N87 cells transfected with the HMGCS1-expressing construct were larger than those of control cells (Figure 3A, left). On day 29 postinjection, mRNA levels of HMGCS1, Oct4, and SOX-2 in the subcutaneous tumors excised from the sacrificed mice were detected by quantitative real-time PCR. The results showed that mRNA levels of Oct4 and SOX-2 in the xenografted tumors of mice injected with NCI-N87 cells were augmented by HMGCS1 overexpression (Figure 3A, right). Reciprocally, the xenografted tumor sizes of NCI-N87 cells infected with lentiviruses expressing the siRNA against HMGCS1 were reduced compared with those of control cells after subcutaneous injection into nude mice (Figure 3B, left). The mRNA levels of Oct4 and SOX-2 in the excised tumors were decreased by HMGCS1 knockdown (Figure 3B, right).

Moreover, the AGS cells transfected with the HMGCS1-expressing construct were also intravenously injected into lateral tail vein of nonobese diabetic severe-combined immunodeficiency (NOD-SCID) mice. After 13 weeks, the mice were sacrificed for evaluation of metastatic nodules in lungs (Figure 3C, left). The results showed that the number of metastatic nodules in lungs was significantly increased in mice injected with AGS cells transfected with the HMGCS1-expressing construct compared with those injected with control cells (Figure 3C, right).

### 2.4. HMGCS1 Knockout Suppresses Growth and Progression of Gastric Cancer Cells

The Clustered Regularly Interspaced Short Palindromic Repeats (CRISPR)-associated protein-9 nuclease (CRISPR-Cas9) system, a key technology for genome editing in cellular and animal models [21], was also employed to assess the effect of HMGCS1 knockout on growth and progression of gastric cancer cells in the present study. A short guide RNA (sgRNA) targeting the human HMGCS1 gene had been designed and used to establish two single cell-derived AGS cell clones with HMGCS1 knockout. The knockout of HMGCS1 in these clones had been confirmed by Western blot analysis (Figure 4A) and sequencing of the PCR fragments of their genomic DNAs (Appendix A).

The CRISPR-Cas9-mediated HMGCS1 knockout of AGS cells resulted in reducing protein levels of Oct4 and SOX-2 by Western blot analysis compared with control cells (Figure 4A). Cell numbers, determined by trypan blue assay (Figure 4B), and migration and invasion (Figure 4C) in AGS cells, were suppressed after HMGCS1 knockout. Accordingly, the xenografted tumor sizes of AGS cells with CRISPR-Cas9-mediated HMGCS1 knockout (Figure 4D, upper) were reduced compared with those of control cells after subcutaneous injection into nude mice. The mRNA levels of Oct4 and SOX-2 in their excised tumors were decreased by HMGCS1 knockout (Figure 4D, lower). The data also showed that the number of metastatic nodules in lungs was significantly decreased in NOD-SCID mice injected with AGS cells with HMGCS1 knockout by intravenous injection into the lateral tail vein compared with those injected with control cells (Figure 4E).

### 2.5. HMGCS1 Exerts Nonmetabolic Functions in Regulation of Gastric Cancer Progression

Next, we questioned whether HMGCS1 contributes to gastric cancer progression in a nonmetabolic manner besides the mevalonate pathway. Based on the results of the MTT assay (Appendix A), mevalonolactone treatment promoted cell growth in AGS cells, but this increment was attenuated by HMGCS1 knockout. These results suggested that HMGCS1 might also have a nonmetabolic function in gastric tumorigenesis.

It has been corroborated that statins inhibit the activity of the rate-limiting enzyme HMGCR in the mevalonate pathway and are associated with reduced cancer-related mortality [8,9]. When co-treated with statin, dipyridamole enhances the statin-induced inhibition of the mevalonate pathway [13]. Therefore, the combination treatment of statin and dipyridamole was used to block mevalonate pathway in this study. Western blot analysis was employed to evaluate the HMGCS1 expression levels in AGS, KATO III, and NCI-N87 cells after drug treatment. Combination treatment with lovastatin and dipyridamole induced a greater downregulation of HMGCS1 protein expression than the individual drugs (Figure 5A). To further check the efficiency of blocking the mevalonate pathway by the combination treatment, we detected mRNA levels of HMGCS1, HMGCR, and Ras homolog family member B (RhoB) by quantitative real-time PCR. The results showed that the mRNA expression of HMGCS1 and HMGCR in AGS and KATO III cells was reduced by combination treatment, whereas mRNA expression of HMGCS1 and HMGCR of RhoB was increased (Appendix A). Additionally, RhoA expression in nonmembrane fractions of AGS cells was upregulated by combination treatment, going along with the downregulation of those in membrane fractions (Appendix A). The suppression of tumorsphere formation in KATO III and NCI-N87 cells by this combination treatment was partially reversed after mevalonolactone treatment (Appendix A).

To check for the possibility of HMGCS1-mediated nonmetabolic functions in gastric carcinogenesis, exogenous HMGCS1 was expressed in AGS, KATO III, and NCI-N87 cells after the blockade of the mevalonate pathway by combination treatment with lovastatin and dipyridamole. An MTT assay showed that the cell growth reduction of AGS, KATO III, and NCI-N87 cells caused by this combination treatment was restored, at least partially, after transfection with the HMGCS1-expressing construct (Figure 5B). Using tumorsphere formation (Figure 5C), migration, invasion, and colony formation (Figure 5D) assays, the progression inhibition of KATO III and NCI-N87 cells caused by this combination treatment was also partially reversed after transfection with the HMGCS1-expressing construct. In contrast, the MTT assay showed that the cell growth reduction caused by combination treatment with lovastatin and dipyridamole was exacerbated by infection with lentiviruses expressing siRNAs against HMGCS1 in NCI-N87 cells (Appendix A, upper) or knockout of HMGCS1 in AGS cells (Appendix A, middle). The decrease in tumorsphere formation in NCI-N87 cells after this combination treatment further deteriorated after infection with lentiviruses expressing siRNAs against HMGCS1 (Appendix A, lower).

### 2.6. Nuclear HMGCS1 Translocation of Gastric Cancer Cells is Induced by Serum Deprivation

It has been reported that the nuclear translocation of glycolytic enzyme pyruvate kinase M2 (PKM2) can be induced under various conditions and then exert its nonmetabolic functions to contribute to tumorigenesis [22,23]. Under various cellular stresses, cholesterogenic enzymes HMGCR [24] and NAD(P)H steroid dehydrogenase-like (NSDHL) [25] can translocate to other cell organelles. Additionally, Figure 2A shows that HMGCS1 induced transcriptional upregulation of Oct4 and SOX-2 in gastric cancer cells. Therefore, under stress conditions, whether the translocation of cytosolic HMGCS1 into nuclei regulates gastric cancer progression through a nonmetabolic manner was analyzed.

To assess this possibility, we first checked for HMGCS1 in the nuclei of gastric cancer cells. Western blot analyses showed that HMGCS1 was detected in both cytosolic and nuclear extracts of AGS and KATO III cells cultured in complete media containing 10% fetal bovine serum (FBS) (Figure 6A). To further confirm the existence of nuclear HMGCS1 in gastric cancer cells, immunoprecipitation was also applied to the nuclear and cytosolic fractions of AGS cell lysates. Endogenous HMGCS1 was immunoprecipitated from both cytosolic and nuclear extracts by anti-HMGCS1 antibody (Figure 6B). HMGCS1 expression in tumors and adjacent normal tissues from gastric cancer patients was examined using immunohistochemical analysis. The results showed that HMGCS1 levels in tumors were higher than those in the adjacent nontumor tissues (Appendix A). Additionally, HMGCS1 was detected in both the nuclei and cytoplasm of gastric cancer tissues (Appendix A).

Additionally, reporter gene assays were performed to check whether HMGCS1 activates Oct4 and SOX-2 expression, as demonstrated (Figure 2A,B) through the enhancing activities of their promoters. After co-transfection with the HMGCS1-expressing construct and reporter plasmids containing promoters of either Oct4 or SOX-2 into AGS and NCI-N87 cells, reporter gene activities were enhanced by HMGCS1 (Figure 6C, upper). In contrast, activities of reporter genes were inhibited by HMGCS1 knockdown after co-transfection with siRNA vectors against HMGCS1 and reporter plasmids containing promoters of either Oct4 or SOX-2 into AGS cells (Figure 6C, lower). These results imply that nuclear HMGCS1 in the context of living cells could bind to the chromosomal DNAs of Oct4 and SOX-2 promoters to induce promoter activities. The chromatin immunoprecipitation (ChIP) assay was performed to examine the DNA-binding ability of HMGCS1 on Oct4 and SOX-2 promoters in AGS cells using anti-HMGCS1 antibody. The ChIP assays clearly showed that HMGCS1 bound to Oct4 and SOX-2 promoters in the chromosomal DNAs of AGS cells (Figure 6D). The binding of HMGCS1 to Oct4 and SOX-2 promoters was specific, as no chromosomal DNA fragment in the coding sequences of Oct4 and SOX-2 was immunoprecipitated with anti-HMGCS1 antibody in AGS cells.

Interestingly, the contents of cytosolic and nuclear HMGCS1s of gastric cancer cells cultured in stress media containing 0.5% FBS were significantly elevated compared with those of cells cultured in complete media containing 10% FBS (Figure 6A). To further explore the dynamics of nuclear translocation of HMGCS1 under serum deprivation, subcellular localization of HMGCS1 was visualized by time-lapsed epi-fluorescence and differential interference contrast (DIC) microscopy after transfection with a plasmid expressing enhanced green fluorescent protein (EGFP)-tagged HMGCS1 into KATO III cells. EGFP-HMGCS1 fusion protein was detected in the transfected KATO III cells by Western blot analysis (Figure 6E). The fluorescence signals of EGFP presented primarily in cytosolic areas in EGFP‒HMGCS1 fusion protein-expressing KATO III cells cultivated in 10% FBS conditions (Figure 6F, middle), resulting in a ratio of nuclear to cytosolic EGFP intensity of below 1 (Figure 6G, blue triangle). In contrast, the EGFP signals translocated into the nuclei when the culture was changed to 0.5% FBS in EGFP‒HMGCS1 fusion protein-expressing KATO III cells (Figure 6F, lower). Quantitation results revealed that the ratio of nuclear to cytosolic EGFP intensity rose gradually in a time-dependent manner (Figure 6G, red circle). After transfecting KATO III cells with EGFP-expressing plasmid as a control group, the transfected KATO III cells displayed an equal distribution of EGFP in both nuclear and cytosolic areas in 0.5% of FBS (Figure 6F, upper and Figure 6G, green circle).

### 2.7. HMGCS1 Enhances the Integrated Stress Response (ISR) Pathway and Interacts with Protein Kinase RNA-Like Endoplasmic Reticulum (ER) Kinase (PERK), One of the Major ER Stress Transducers

The integrated stress response (ISR) pathway is activated in response to diverse stress stimuli in eukaryotic cells and includes the phosphorylation of eukaryotic translation initiation factor 2 alpha (eIF2α), the core event of ISR, and the upregulation of activating transcription factor 4 (ATF4), a key effector of ISR [26,27]. The increase in nuclear HMGCS1 in gastric cancer cells caused by serum deprivation (Figure 6) prompted us to investigate whether HMGCS1 is involved in modulating the ISR pathway. Immunoblots showed that serum deprivation in KATO III and NCI-N87 cells induced higher levels of HMGCS1, phosphorylated eIF2α (p-eIF2α), and ATF4 (Figure 7A). Levels of p-eIF2α were also elevated in AGS, KATO III, and NCI-N87 cells either treated with 2.5 μg/mL tunicamycin, an inducer of ER stress, for 6 h or transfected with the HMGCS1-expressing construct (Figure 7B).

Moreover, expression levels of eIF2α kinases such as general control nonderepressible 2 (GCN2) and PERK in AGS, KATO III, and NCI-N87 cells were not affected after transfection with the HMGCS1-expressing construct (Figure 7C). However, levels of the phosphorylated PERK (p-PERK) and ATF4 in these cells were enhanced after transfecting the HMGCS1-expressing construct (Figure 7C). The mRNA expression of ER stress markers such as unspliced (X-box binding protein (XBP) 1u) and spliced (XBP1s) forms of XBP1, total XBP1, ATF4, and C/EBP homologous protein (CHOP) in NCI-N87 cells was also enhanced after transfecting the HMGCS1-expressing construct (Appendix A).

PERK is an ER-resident protein [28] and its activation is modulated by protein‒protein interaction between PERK and 78-kDa glucose-regulated protein (GRP78) [26,27]. Because almost all enzymes involved in the synthesis of cholesterol localize in the ER [6], whether HMGCS1 co-localizes and interacts with PERK was analyzed by immunofluorescence staining as well as subsequent analyses of confocal microscopy and co-immunoprecipitation. Immunofluorescence staining showed that HMGCS1 mainly localized in the cytoplasm of AGS cells and co-localized with PERK (Figure 7D). Notably, there were faint but obvious staining specks of HMGCS1 in the nuclei of AGS cells (Figure 7D). Co-immunoprecipitation assay of cell lysates of AGS cells showed that PERK and HMGCS1 were co-immunoprecipitated with anti-HMGCS1 antibody (Figure 7E), which coincides with the observation of their colocalization in immunofluorescence staining (Figure 7D).

It has been found that an increase in ATF4 and CHOP levels precedes cell death under mild ER stress, indicating that mild ER stress can be activated without necessitating cell death [29,30]. To understand the effect of HMGCS1 on the growth and progression of gastric cancer cells under mild ER stress, AGS cells with HMGCS1 knockout and AGS cells infected with lentiviruses expressing siRNAs against HMGCS1 were seeded for cell growth (MTT), migration, and invasion assays. Cells were treated with 2.5 μg/mL tunicamycin for 6 h to elicit mild ER stress before assessment. The mild ER stress led to a modest, but statistically significant, improvement in the growth, migration, and invasion of AGS (Figure 7F), and an increase in NCI-N87 cells (Appendix A). Under mild ER stress, knockout and knockdown of HMGCS1 reversed the tunicamycin-mediated increase in the growth and progression of AGS (Figure 7F) and NCI-N87 (Appendix A) cells, respectively. However, a longer tunicamycin-treatment period added at the beginning of the experiments suppressed the growth and progression of AGS cells with and without HMGCS1 knockdown (Appendix A).

## 3. Discussion

The role and regulatory mechanism of HMGCS1 in cancer development and progression are complicated and not yet fully understood. HMGCS1 is a potential regulatory node of the mevalonate pathway involved in tumorigenesis [2,3,4]. This study indicated that HMGCS1 had both metabolic and nonmetabolic functions in potentiating gastric cancer progression. Furthermore, levels of HMGCS1 in gastric cancer cells could be induced by serum deprivation, and the increase in HMGCS1 expression induced the ISR pathway. To our knowledge, this is the first report regarding the nonmetabolic function of HMGCS1 and the link between HMGCS1 and the ISR pathway in controlling cancer progression (Figure 7G).

Herein, we also showed for the first time that the nuclear translocation of HMGCS1 in gastric cancer cells was induced after serum deprivation. An intriguingly similar situation is that expression of glycolytic enzyme PKM2 is upregulated in certain tumors and associated with poor prognosis [22,23]. Under various conditions, PKM2 is induced for nuclear translocation and then exerts its nonglycolytic functions. Nuclear PKM2 can act as a kinase for STAT3 phosphorylation and as a transcriptional co-activator by interacting with Oct4 as well as phosphorylated β-catenin, which in turn activates the transcription of target genes to promote tumorigenesis [22,23]. Thus, both the metabolic and nonmetabolic functions of PKM2 promote tumorigenesis [23].

Our study suggested that HMGCS1 could contribute to gastric cancer progression in a nonmetabolic manner (Figure 5, Appendix A, and Appendix A). However, the regulatory mechanism of HMGCS1-mediated nonmetabolic function on enhancing gastric tumorigenesis is still unclear. The data showed that HMGCS1 could localize into nuclei (Figure 6A,B), upregulate Oct4 and SOX-2 levels (Figure 2A,B), and bind to Oct4 and SOX-2 promoters (Figure 6D) in gastric cancer cells. Thus, we believe nuclear HMGCS1 could promote gastric cancer progression by directly or indirectly binding to pluripotency genes and inducing their expression. Furthermore, ER stress inducer tunicamycin augmented HMGCS1 expression in gastric cancer cells, and HMGCS1 overexpression raised levels of p-eIF2α (Figure 7B), p-PERK, ATF4 (Figure 7C), and ER stress markers (Appendix A). Therefore, HMGCS1 can activate the ISR pathway in gastric cancer cells.

This study also demonstrated that the slight enhancement of the growth and progression of gastric cancer cells after tunicamycin-induced mild ER stress was HMGCS1-dependent (Figure 7F and Appendix A). This is correlated with the report that cells continuously proliferate under mild ER stress following treatment with a low concentration of tunicamycin in mouse embryonic fibroblasts [29]. The p-eIF2α level is induced after tunicamycin treatment for 1 to 2 h, followed by an increase in ATF4 and CHOP expression [30]. However, cleavage of caspase 3 and PARP, involved in the pro-apoptotic pathway, is detected after tunicamycin treatment for 24 h [30].

Our data also proved that HMGCS1 interacted with PERK (Figure 7E) and induced phosphorylation of PERK (Figure 7C) in gastric cancer cells. HMGCS1 may activate the ISR pathway by associating with PERK, which, in turn, augments gastric cancer progression in a nonmetabolic manner. In addition, the ISR pathway can be triggered by oncogene activation in cancer cells [31,32]. For example, the activation of c-Myc had been demonstrated to activate the PERK/eIF2α/ATF4 signaling axis of the ISR pathway, which is indispensable for c-Myc-induced transformation and tumorigenesis [31]. c-Myc-induced ATF4 inhibits apoptosis and promotes the survival of mouse embryonic fibroblasts [33]. Using a similar mechanism, HMGCS1 could potentiate the ISR pathway and promote gastric cancer progression via a nonmetabolic pathway.

HMGCS1 expression is regulated by SREBP2 [6], ACBP [11], mutant p53 [12], and ferrous as well as ferric irons [16]. Our data showed that HMGCS1 levels were also upregulated by serum deprivation (Figure 7A) and ER stress inducer tunicamycin (Figure 7B). In a mouse model, treatment with tunicamycin activates SREBP isoforms SREBP1a and SREBP1c in liver tissues [34]. However, there is little effect on the tunicamycin-induced activation of SREBP2, which promotes HMGCS1 expression and plays a major role in cholesterol biosynthesis [34]. Furthermore, there are several post-translational modifications on HMGCS1, including phosphorylation [35], acetylation [36], ubiquitination [37,38,39], and sumoylation [7]. The biological significance of these modifications on HMGCS1 is still largely elusive at present. HMGCS1 is deacetylated in the cytoplasm by Sirtuin 1 (SIRT1), located in both the cytoplasm and nuclei [40]. Data from this study also raise the possibility that HMGCS1 could be deacetylated by nuclear SIRT1. We will further dissect the biological functions of modifications on HMGCS1, especially nuclear HMGCS1.

Owing to the feedback response triggered by the inhibition of the mevalonate pathway after statin treatment, HMGCR and HMGCS1 genes are upregulated and suppress the anticancer potential of statins [11]. Impeding this feedback response by dipyridamole, the combination of FDA-approved drugs statins and dipyridamole may potentiate the anticancer efficacy of statins for treatment of patients with acute myelogenous leukemia and multiple myeloma [13]. Furthermore, the ISR pathway plays a dual role, both enhancing survival and promoting the death of cancer cells, so pharmacological modulation of the ISR pathway is a potential therapeutic strategy [26,27]. Our results showed that HMGCS1 augmented levels of components of the ISR pathway including p-eIF2α, ATF4, and p-PERK (Figure 7B,C). Therefore, targeting components of the ISR pathway in combination with statin and dipyridamole treatment may be a novel therapeutic strategy for the treatment of gastric cancer patients with higher levels of HMGCS1 and these components in the future.

## 4. Materials and Methods

### 4.1. Plasmids and Plasmid Construction

Constructs pHMGCS1-Flag [40] and pEGFP-HMGCS1 contain the cDNAs encoding HMGCS1 with C-terminal Flag and N-terminal EGFP tags, respectively. Luciferase reporter plasmids Oct4-Luc and SOX-2-Luc contain Oct4 and SOX-2 promoters, respectively [41]. For knockdown of the endogenous HMGCS1, the target sequences listed in Appendix A were constructed in the siRNA vector pLKO.1 as described [42]. The constructs used in the present study were verified by sequencing.

### 4.2. Cell Culture and Transfection

Human stomach adenocarcinoma AGS (adherent cells with wild-type TP53), NUGC-3 (adherent cells with mutant TP53), KATO III (adherent and suspension cells with deleted TP53), SNU-16 (suspension cells with mutant TP53), and NCI-N87 (adherent cells with mutant TP53) cells were obtained from the Bioresource Collection and Research Center (Food Industry Research and Development Institute, Hsinchu, Taiwan). Cells were cultured in RPMI 1640 medium containing 10% FBS. Transfection of plasmids was performed using either electroporation or transfection reagents such as LipofectamineTM 2000 (Invitrogen, Carlsbad, CA, USA) and PolyJetTM (SignaGen, Frederick, MD, USA). Cells were seeded onto six-well plates and then transfected for subsequent luciferase reporter gene assay. After transfection for two days, luciferase activity was measured and then normalized [43]. Lovastatin, dipyridamole, and tunicamycin (Sigma-Aldrich, St. Louis, MO, USA) in dimethyl sulfoxide (DMSO), and mevalonolactone (Sigma-Aldrich) in phosphate-buffered saline (PBS) or an equal volume of vehicles, were added to the media for drug treatment.

For the production of lentiviruses expressing siRNAs, HEK 293T cells (3 × 10^6^/10-cm-diameter dish) were seeded for 24 h and then transfected with vectors including pLKO.1shRNA, pCMVdeltaR8.91, and pMD.G using Lipofectamine^TM^ 2000 (Invitrogen). After 24 and 48 h post-transfection, the collected culture medium was clarified by centrifugation and the supernatant was subsequently filtered using a 0.45-μm pore-size filter at 4 °C.

### 4.3. The CRISPR-Cas9-Mediated HMGCS1 Knockout of AGS Gastric Cancer Cells

For the establishment of stable AGS cells with CRISPR-Cas9-mediated HMGCS1 knockout, oligonucleotides with the sgRNA sequence targeting HMGCS1 were synthesized according to CRISPR design (http://crispr.mit.edu/). Their sequences are listed in Appendix A. Primer pairs were annealed and then cloned into the lentivirus transfer vector lentiCRISPRv2, digested by restriction enzyme BsmBI to produce the recombinant lentiviral plasmid. The recombinant lentiviral plasmid expressing a sgRNA was then sequenced to validate the insertion of the sgRNA sequence. The validated plasmid was co-transfected with packaging plasmids pCMVdeltaR8.91 and pMD.G into HEK 293T cells to produce lentiviruses expressing the sgRNA for subsequent transduction. To obtain single-cell clones of HMGCS1-knockout cells, the infected AGS cells were cultured under puromycin selection (1 μg/mL) through serial dilutions and then screened by Western blot analysis with anti-HMGCS1 antibody. For the detection of insertion and deletions (indels), genomic DNAs of AGS cells with HMGCS1 knockout (HMGCS1 KO #BF4 and #FE4 cells) were extracted for PCR amplification of the modified region. PCR products of the modified region were then cloned into a T&A vector for sequencing. The lentiCRISPR v2 empty vector was also used to generate recombinant lentiviruses for transduction into AGS cells as negative control cells.

### 4.4. Quantitative Real-Time PCR Analysis

After extraction of total RNAs by Trizol reagent (Invitrogen), cDNAs were synthesized using Moloney murine leukemia virus reverse transcriptase (New England BioLabs, Beverly, MA, USA) with an oligo (dT)_18_ primer. Then, cDNAs were amplified with primers by an ABI StepOne Plus system with SYBR Green Master Mix for mRNA detection according to the manufacturer’s protocol (Applied Biosystems, Thermo Fisher Scientific, Warrington, UK) as described previously [43,44]. All primers used for PCR are listed in Appendix A. The relative quantification of mRNA levels was calculated by use of the 2^−ΔΔCT^ values and then normalized with those of GAPDH. The data are representative of the mean values and standard deviations of at least three independent experiments.

### 4.5. Western Blot Analysis

For the preparation of whole-cell extracts, cells were washed with PBS and then lysed in NETN buffer (50 mM Tris-HCl (pH 7.9), 150 mM NaCl, 0.5 mM EDTA, and 0.5% NP40) containing protease and phosphatase inhibitors as described above. Proteins of cell extracts were denatured in sample buffer and further analyzed by SDS-PAGE as described before [45]. Subsequently, Western blot analysis was performed using anti-HMGCS1 (GeneTex, Irvine, CA, USA and Santa Cruz, CA, USA), anti-Oct4 (GeneTex), anti-SOX-2 (GeneTex), anti-EGFP (GeneTex), anti-eIF2α (GeneTex), anti-p-eIF2α (Cell Signaling), anti-PERK (GeneTex), anti-p-PERK (GeneTex), anti-ATF4 (GeneTex), anti-GCN2 (GeneTex), anti-B23 (Santa Cruz), and anti-glyceraldehyde-3-phosphate dehydrogenase (GAPDH, GeneTex) antibodies.

### 4.6. Subcellular Fractionation

To prepare the nuclear and cytosolic extracts, cell pellets were lysed in a hypotonic buffer (1 mM MgCl_2_, 10 mM KCl, 20 mM HEPES (pH 7.4), 0.5% Nonidet P-40, 0.5 mM dithiothreitol, 1 mM phenylmethylsulfonyl fluoride, 10 μg/mL leupeptin, 10 μg/mL aprotinin, and 100 mM sodium fluoride) on ice for 30 min. After centrifugation at 4000× *g* at 4 °C for 10 min, the supernatants removed from the pellets were treated as cytosolic extracts. Pellets of nuclei were further resuspended and lysed in a high-salt buffer (0.4 M NaCl, 1 mM MgCl_2_, 10 mM KCl, 20 mM HEPES (pH 7.4), 0.5 mM dithiothreitol, and protease and phosphatase inhibitors, as described above) at 4 °C for 30 min. After centrifugation, the recovered supernatants were treated as nuclear extracts.

### 4.7. Co-immunoprecipitation

Whole-cell extracts of cells (1 × 10^5^) were prepared as described above. To prepare the slurry of antibody-conjugated protein A-Sepharose, 1 μg of anti-HMGCS1 antibody (sc-166763, Santa Cruz) or rabbit IgG (I5006, Sigma-Aldrich), 50 μL of a 50% (vol/vol) slurry of protein A-Sepharose, and 450 μL of NETN buffer were incubated overnight at 4 °C under gentle rotation. Then whole-cell extracts (250 μg) and antibody-conjugated protein A-Sepharose were mixed overnight at 4 °C under rotary agitation. The immunoprecipitated proteins were eluted from the beads by boiling samples in sample buffer for Western blot analysis using anti-HMGCS1 (GTX112346, GeneTex) and anti-PERK (GTX81222, GeneTex) antibodies.

### 4.8. Cell Growth and Viability Assays

To evaluate cell growth and viability, the treated cells were seeded onto six-well plates and then counted by trypan blue exclusion method. Additionally, MTT assay was carried out and then determined using a microplate ELISA reader (TECAN Infinite 200, Männedorf, Switzerland) after incubation for 24 or 48 h [46].

### 4.9. Colony Formation, Migration, Invasion, and Tumorsphere Formation Assays

To assay the ability of anchorage-independent growth, the treated cells were seeded in soft agar for subsequent colony formation assay [43]. Fourteen days later, numbers of colonies larger than 0.1 mm in diameter were counted from 10 random fields under the microscope. As described [43], the migration and invasion of the treated cells were examined in 24-well plates for 12 and 20 h, respectively. The migrated or invaded cell numbers were counted from 10 random fields under the microscope. To survey tumorsphere formation, the treated cells seeded onto 96-well ultra-low-attachment plates (Corning) were cultured in a stem cell-selective medium [47]. A count of spheres larger than 50 μm was made under the microscope seven days later.

### 4.10. ChIP Assay

To cross-link DNA and protein, cells were incubated with formaldehyde to a final concentration of 1% at room temperature for 15 min. The reaction was stopped by adding glycine (0.125 M final concentration) at room temperature for 5 min. Then, cells were lysed in 500 μL of NETN buffer (50 mM Tris-HCl (pH 7.9), 150 mM NaCl, 0.5 mM EDTA, and 0.5% NP40) with inhibitors of proteases (1 mM phenylmethylsulfonyl fluoride, 10 μg/mL aprotinin, and 10 μg/mL leupeptin) and phosphatases (100 mM sodium fluoride). After sonication with a microtip and subsequent centrifugation, the supernatant was immunoprecipitated with a 50% (vol/vol) slurry of anti-HMGCS1 antibody (sc-166763, Santa Cruz) or rabbit IgG (I5006, Sigma-Aldrich)-conjugated protein A-Sepharose overnight at 4 °C. The immunoprecipitated DNA–protein complexes were eluted with 150 μL of elution buffer (50 mM NaHCO_3_, 1% SDS) twice by vortexing. After adding 30 μg of RNase A, 36 μL of 5 M NaCl, and elution buffer to a final volume of 600 μL, the immunoprecipitates were heated at 67 °C for 5 h to reverse the cross-linking of DNA and protein. Then, DNAs were extracted with phenol and then precipitated by ethanol. In total, 10% of the precipitated DNA was used as a template for real-time PCR using the specific primers as listed in Appendix A to amplify the DNA fragments of Oct4 and SOX-2 promoters. The percentages of immunoprecipitated fragments of promoters in chromosomal DNA were quantified and then normalized to total input DNA.

### 4.11. Time-Lapsed Epi-Fluorescence and DIC Microscopy Imaging

To record time-lapsed images of transportation of EGFP-tagged HMGCS1 in live cells, KATO III cells were inoculated on a 3.5-cm glass-bottom plate (Mettek, Ashland, MA, USA) and then transfected with either EGFP-tagged HMGCS1-expressing plasmid (pEGFP-HMGCS1) or EGFP-only expressing vector (pEGFP-C1). Both phase-contrast and fluorescence images of cells were acquired by using an inverted microscope (IX 71, Olympus, Tokyo, Japan) equipped with an Olympus DP70 digital camera system. EGFP was excited by 460–490 nm wavelength, filtered from a mercury light source. The fluorescent emission of EGFP was collected by a 100× oil immersion objective with a numerical aperture (N.A.) value of 1.40 (Olympus, Japan), and filtered by a WIBA filter cube (515–550 nm). The phase-contrast images were acquired prior to fluorescence imaging for displaying the shapes of cells during the recording period. All images were recorded with DP-BSW software (Olympus, Tokyo, Japan). Images of 10 individual cells were acquired at the indicated time points. To elucidate the alteration of the fluorescence in both the cytoplasm and nuclei, the ratio of the fluorescence intensity in nuclear and cytoplasmic areas was determined using ImageJ software (version 1.49, University of Wisconsin, Madison, WI, USA).

### 4.12. Immunofluorescence Staining and Confocal Microscopy

Cells (1 × 10^5^) seeded on coverslips were fixed with 4% paraformaldehyde at room temperature for 8 min. After blocking with 2% bovine serum albumin and subsequent washing with PBS, the cells were incubated with primary mouse anti-HMGCS1 (sc-166763, Santa Cruz) or rabbit anti-PERK (GTX81222, GeneTex) antibodies, followed by secondary DyLight™ 488-conjugated goat anti-mouse IgG (GTX213111-04, GeneTex) or DyLight™ 594-conjugated goat anti-rabbit IgG (GTX213110-05, GeneTex). The cells were also stained with 4,6-diamidino-2-phenylindole dihydrochloride (DAPI) (Sigma-Aldrich) for nuclear staining. The subcellular localizations of HMGCS1 and PERK were further examined by confocal microscopy (Zeiss LSM880 with Airyscan, Jena, Germany).

### 4.13. In vivo Xenografted Tumorigenicity and Tail Vein Metastasis Assays

Animal experiments in this study were performed in accordance with protocols approved by the institutional ethical committee (Institutional Animal Care and Use Committee of National Yang-Ming University). The animal ethics approval codes are 1060113, 1070501, and 1090107. In brief, five-week-old BALB/c nu/nu mice (National Laboratory Animal Center, Taipei, Taiwan) were subcutaneously injected with the treated cells for xenografted tumorigenicity assay as described [39]. Both the length (L) and width (W) of the xenografts were measured with calipers every two days. The tumor volume (V) was estimated using the formula V = (L × W^2^)/2. Five-week-old male nonobese diabetic severe-combined immunodeficiency (NOD-SCID) mice (National Laboratory Animal Center, Taipei, Taiwan) were injected with the treated cells by tail vein injection for metastasis assay [43]. After sacrificing, lung samples were collected and rinsed by PBS. The metastatic burden was assessed by counting the numbers of white nodules on the surface of lungs under a dissecting microscope.

### 4.14. Surgical Samples

Gastric adenocarcinoma tissues from gastric cancer patients were collected at the Department of Surgery, Taipei Veterans General Hospital. Before surgery, none of these patients were undergoing chemotherapy or radiotherapy. Informed consent was obtained from each patient before collection. An analysis of tissue specimens was performed according to a protocol approved by the Institutional Review Board of Taipei Veterans General Hospital. The ethics codes are TPEVGH No. 2017-04-004AC, 2017-12-018BC, and 2020-01-011AC.

### 4.15. Statistical Analyses

Statistical analysis was performed using an independent Student’s *t*-test for the simple comparison of two groups. A difference in the results was considered to be statistically significant when the *P* value was less than 0.05.

## 5. Conclusions

In summary, our results reveal that HMGCS1 exerts both metabolic and nonmetabolic functions in gastric cancer progression. HMGCS1 translocates into the nuclei of gastric cancer cells under stress conditions and binds to and activates Oct4 and SOX-2 promoters. HMGCS1 also interacts with PERK and induces the ISR pathway, suggesting that HMGCS1 and the ISR pathway may be promising targets for therapy in gastric cancer.

## Figures and Tables

**Figure 1 cancers-12-01088-f001:**
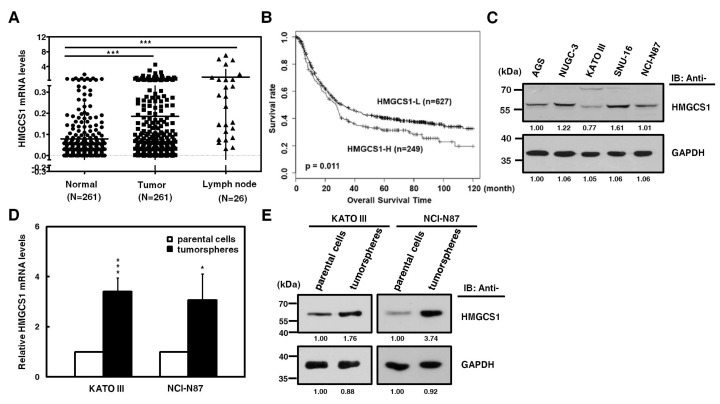
3-hydroxy-3-methylglutaryl-CoA synthase 1 (HMGCS1) expression is upregulated in stomach adenocarcinoma samples of patients and tumorspheres of gastric cancer cells. (**A**) Levels of HMGCS1 mRNA in tumors, the adjacent nontumor tissues, and lymph node tumor sample pairs from gastric cancer patients (HMGCS1 in tumors and the adjacent nontumor tissues, *n* = 261; HMGCS1 in lymph node tumor samples, *n* = 26) were examined using quantitative real-time PCR analysis. HMGCS1 mRNA levels in the gastric cancer tissues or lymph node tumor samples were compared with those of the corresponding adjacent normal tissues. Mean ± SD. *** *p* < 0.001. (**B**) The Kaplan‒Meier survival plot of gastric cancer patients with higher (HMGCS1-H, *n* = 249) or lower (HMGCS1-L, *n* = 627) levels of HMGCS1 mRNA. *p* = 0.011. (**C**) Whole-cell extracts of gastric cancer cells including AGS, NUGC-3, KATO III, SNU-16, and NCI-N87 cells were prepared for Western blot analysis using anti-HMGCS1 and anti-glyceraldehyde-3-phosphate dehydrogenase (GAPDH) antibodies. (**D****, E**) KATO III and NCI-N87 cells were seeded onto ultra-low attachment plates under stem cell-selective conditions for the subsequent formation assay of tumorspheres. The transcript levels of HMGCS1 in parental cells and tumorspheres of KATO III and NCI-N87 cells were measured by quantitative real-time PCR and then normalized to GAPDH (**D**). Mean ± SD (*n* = 3). * *p* < 0.05; *** *p* < 0.001. Whole-cell extracts of parental cells and tumorspheres of KATO III and NCI-N87 cells were prepared for Western blot analysis using anti-HMGCS1 and anti-GAPDH antibodies (**E**).

**Figure 2 cancers-12-01088-f002:**
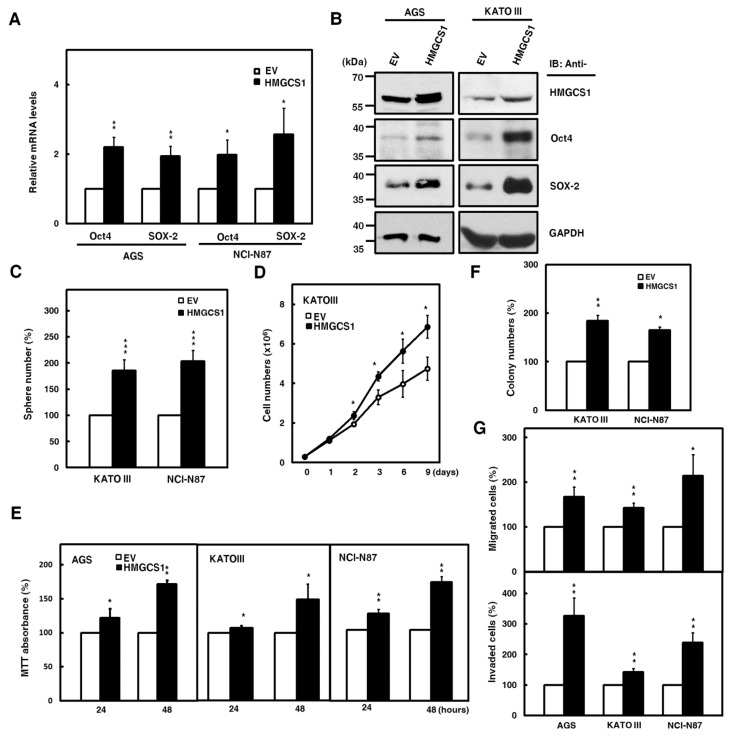
HMGCS1 elevates the levels of pluripotency genes Oct4 and SOX-2 and contributes to progression in gastric cancer cells. (**A**,**B**) AGS, NCI-N87, and KATO III cells were transfected with the HMGCS1-expressing plasmid construct (HMGCS1) or empty vector (EV) for 48 h. The transcript levels of Oct4 and SOX-2 in the transfected AGS and NCI-N87 cells were determined by quantitative real-time PCR (A). Mean ± SD (*n* = 3). Whole-cell extracts of the transfected AGS and KATO III cells were prepared for Western blot analysis using anti-HMGCS1, anti-Oct4, anti-SOX-2, and anti-GAPDH antibodies (B). (**C**) KATO III and NCI-N87 cells transfected with the HMGCS1-expressing construct or empty vector for 48 h were seeded for formation assay of tumorspheres. Mean ± SD (*n* = 3). (**D**) KATO III cells transfected with the HMGCS1-expressing construct or empty vector were seeded for cell counting by trypan blue exclusion. Mean ± SD (*n* = 3). (**E**) AGS, KATO III, and NCI-N87 cells transfected with the HMGCS1-expressing construct or empty vector were seeded for 3-(4,5-dimethyl-2-thiazolyl)-2,5-diphenyl tetrazolium bromide (MTT) assay. Mean ± SD (*n* = 3). (**F**) The transfected KATO III and NCI-N87 cells from (**E**) were seeded for colony formation assay. Mean ± SD (*n* = 3). (**G**) The transfected cells from (**E**) were also used for migration (upper) and invasion (lower) assays. Mean ± SD (*n* = 3). *, *p* < 0.05; **, *p* < 0.01; ***, *p* < 0.001.

**Figure 3 cancers-12-01088-f003:**
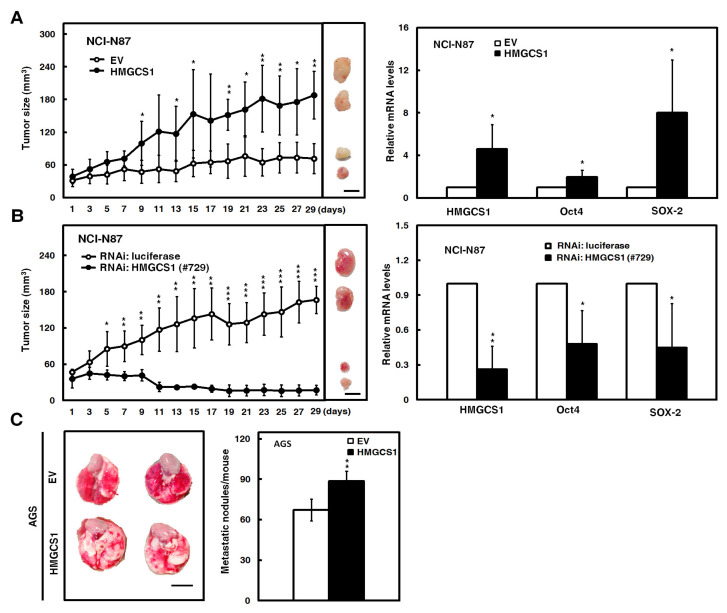
HMGCS1 enhances the tumor growth and lung metastasis of gastric cancer cells. (**A**,**B**) NCI-N87 cells were either transfected with the HMGCS1-expressing construct (HMGCS1) or empty vector (EV) (**A**) or infected with lentiviruses expressing siRNAs against HMGCS1 (#729) or luciferase (**B**). After treatment for 48 h, the treated cells were subcutaneously injected into nude mice (*n* = 4 per group) for measurement of tumor sizes at the time indicated (left). Bar, 0.5 cm. The mice were sacrificed on day 29 and subcutaneous tumors were excised for the detection of HMGCS1, Oct4, and SOX-2 transcripts using quantitative real-time PCR (right). Mean ± SD. (**C**) AGS cells were transfected with the HMGCS1-expressing construct or empty vector for 48 h and then injected into nonobese diabetic severe-combined immunodeficiency (NOD-SCID) mice (*n* = 4 per group) by tail vein injection for measurement of metastatic nodules in lungs. After 13 weeks, the mice were sacrificed and the metastatic nodules in lungs of the treated mice were also counted by gross and microscopic examination. Mean ± SD. *, *p* < 0.05; **, *p* < 0.01; ***, *p* < 0.001.

**Figure 4 cancers-12-01088-f004:**
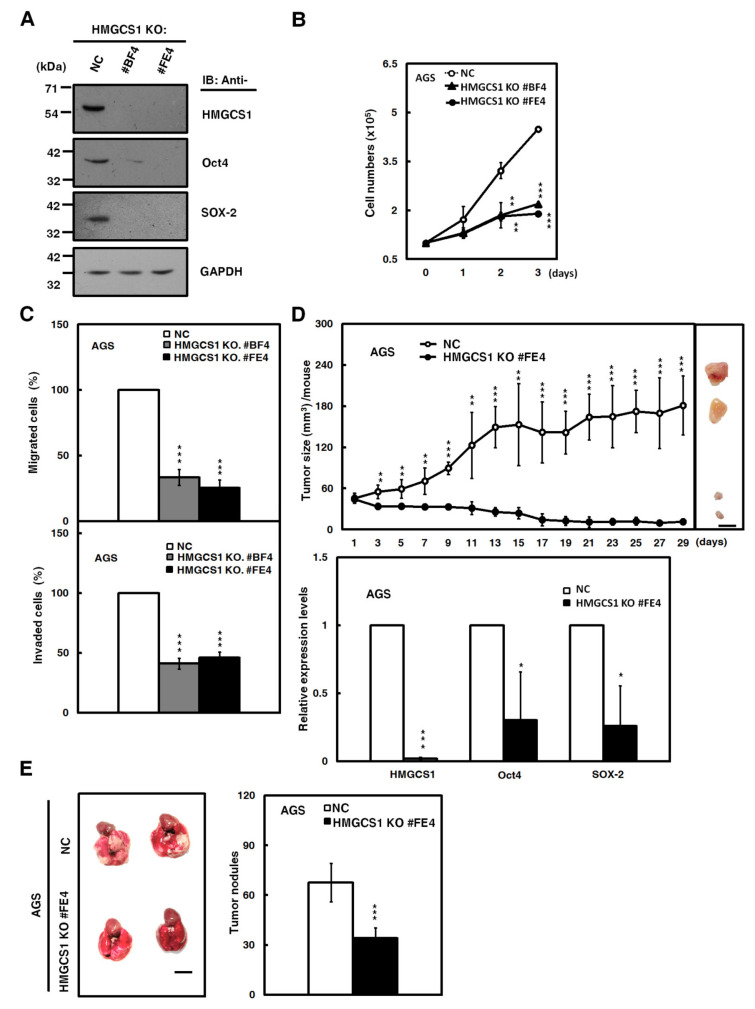
The Clustered Regularly Interspaced Short Palindromic Repeats-associated protein-9 nuclease (CRISPR-Cas9)-mediated HMGCS1 knockout suppresses growth and progression of gastric cancer cells. (**A**) Whole-cell extracts of HMGCS1 knockout (KO) #BF4 and #FE4 cells and negative control (NC) cells were prepared for Western blot analysis using anti-HMGCS1, anti-Oct4, anti-SOX-2, and anti-GAPDH antibodies. (**B**,**C**) HMGCS1 KO #BF4 and #FE4 cells and negative control cells were seeded for cell counting by trypan blue exclusion method (**B**), migration, and invasion (**C**). Mean ± SD (*n* = 3). (**D**) HMGCS1 KO #FE4 cells and negative control cells were subcutaneously injected into nude mice (*n* = 5 per group) for measurement of tumor sizes at the time indicated (upper). Bar, 0.5 cm. The mice were sacrificed on day 29 and then subcutaneous tumors were excised for the detection of HMGCS1, Oct4, and SOX-2 transcripts using quantitative real-time PCR (lower). Mean ± SD. (**E**) HMGCS1 KO #FE4 cells and negative control cells were injected into NOD-SCID mice (*n* = 4 per group) by tail vein injection for measurement of metastatic nodules in lungs. After 11 weeks, the mice were sacrificed and the metastatic nodules in lungs of the treated mice were also counted by gross and microscopic examination. Mean ± SD. *, *p* < 0.05; **, *p* < 0.01; ***, *p* < 0.001.

**Figure 5 cancers-12-01088-f005:**
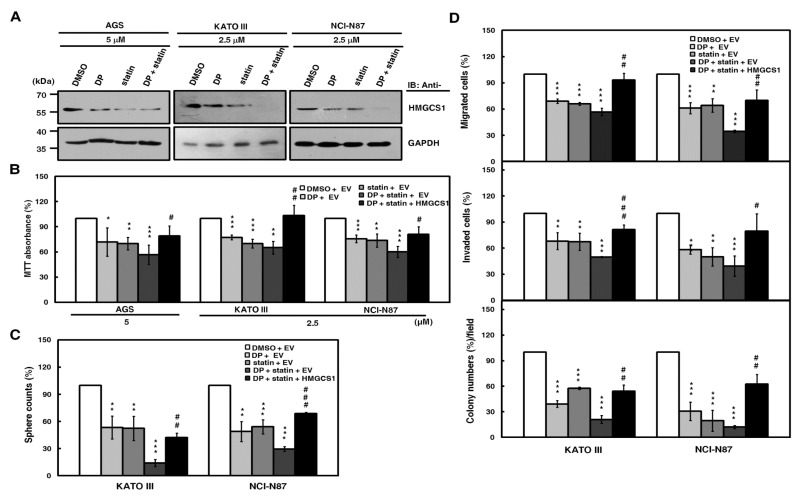
Suppression of growth and progression in gastric cancer cells by the blockade of the mevalonate pathway is partially reversed by HMGCS1 overexpression. (**A**) For the blockade of the mevalonate pathway, AGS, KATO III, and NCI-N87 cells were treated with 2.5 or 5 μM lovastatin (statin) and/or dipyridamole (DP) for 24 h. Then, whole-cell extracts of the treated cells were prepared for Western blot analyses using anti-HMGCS1 and anti-GAPDH antibodies. (**B–D**) AGS, KATO III, and NCI-N87 cells were transfected with the HMGCS1-expressing construct (HMGCS1) or empty vector (EV) for 48 h. Then, the transfected cells were treated with lovastatin and/or dipyridamole for 24 h for the subsequent MTT assay (**B**). The transfected KATO III and NCI-N87 cells were also seeded for assays of tumorsphere formation (**C**), migration, invasion, and colony formation (**D**) in the presence or absence of 2.5 μM of lovastatin and/or dipyridamole. Mean ± SD (*n* = 3). *, *P* < 0.05; **, *P* < 0.01; ***, *p* < 0.001. #, *p* < 0.05; ##, *p* < 0.01; ###, *p* < 0.001.

**Figure 6 cancers-12-01088-f006:**
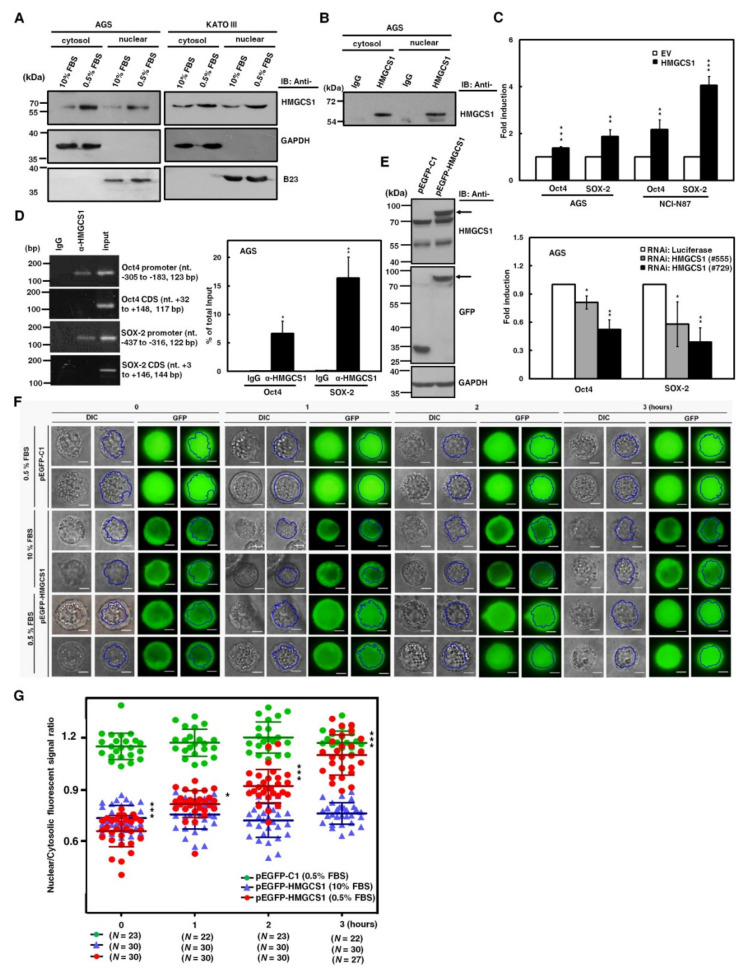
Nuclear HMGCS1 translocation of gastric cancer cells is induced by serum deprivation. (**A**) AGS (left) and KATO III (right) cells were cultured in complete media containing 10% fetal bovine serum (FBS) or under serum deprivation (0.5% FBS) for 24 h. Then cytosolic and nuclear extracts of the treated cells were prepared for Western blot analysis using the anti-HMGCS1, anti-GAPDH, and anti-B23 (a nuclear marker) antibodies. (**B**) Both cytosolic and nuclear extracts of AGS cells were immunoprecipitated with anti-HMGCS1 antibody. The precipitated proteins were then analyzed by Western blot analysis using anti-HMGCS1 antibody. (**C**) After co-transfection with the HMGCS1-expressing construct (HMGCS1) or empty vector (EV) and luciferase reporter plasmids containing promoters of pluripotency genes Oct4 or SOX-2 for 48 h, the transfected AGS and NCI-N87 cells were harvested for reporter gene assay (upper). After co-transfection with siRNA vectors against HMGCS1 (#555 and #729) or luciferase and reporter plasmids containing promoters of pluripotency genes Oct4 or SOX-2 for 48 h, the transfected AGS cells were harvested for reporter gene assay (lower). Mean ± SD (*n* = 3). (**D**) AGS cells were harvested for chromatin immunoprecipitation (ChIP) assay using IgG or anti-HMGCS1 antibody. Then, the immunoprecipitated DNAs were used to amplify the PCR products using the primer pairs specific for the promoters and coding sequences (CDS) of Oct4 and SOX-2 (left). The positions (in relation to the transcription starting site) and lengths of PCR products were also shown. Percentages of the immunoprecipitated DNA fragments of Oct4 and SOX-2 promoters were further quantified by real-time PCR and subsequently normalized to total input DNA (right). Mean ± SD (*n* = 3). (**E**–**G**) After transfecting the enhanced green fluorescent protein (EGFP)-tagged HMGCS1-expressing plasmid (pEGFP-HMGCS1) or EGFP-expressing control vector (pEGFP-C1) into KATO III cells for 48 h, whole-cell extracts of the transfected cells were prepared for Western blot analysis using anti-HMGCS1, anti-GFP, and anti-GAPDH antibodies (**E**). The arrows indicate the EGFP-HMGCS1 fusion proteins. The transfected cells were also cultured in media containing 10% or 0.5% FBS for further examination of the fluorescent signals of EGFP and EGFP-HMGCS1 fusion protein at the indicated times by fluorescent microscopy (**F**). Blue lines indicate cell nuclei. The fluorescent signals were quantified from three randomly selected areas in nuclei and cytoplasm of the transfected cells and then analyzed by ImageJ software. The chart shows ratios of nuclear to cytosolic fluorescent signals in the transfected cells (**G**). Green circles represent cells transfected with EGFP-expressing control vector and cultured in media containing 0.5% FBS, blue triangles represent cells transfected with EGFP-tagged HMGCS1-expressing plasmid and cultured in media containing 10% FBS, and red circles represent cells transfected with EGFP-tagged HMGCS1-expressing plasmid and cultured in media containing 0.5% FBS. Mean ± SD. *, *p* < 0.05; **, *p* < 0.01; ***, *p* < 0.001.

**Figure 7 cancers-12-01088-f007:**
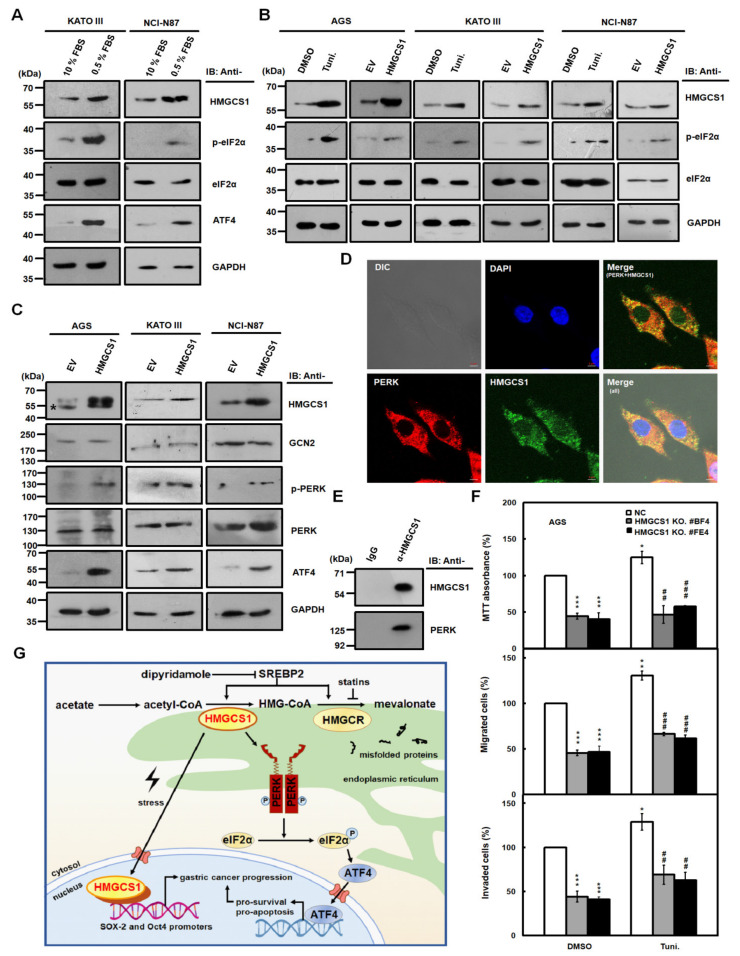
HMGCS1 enhances the integrated stress response (ISR) pathway and interacts with protein kinase RNA-like endoplasmic reticulum kinase (PERK). (**A**–**C**) KATO III and NCI-N87 cells were cultured in complete media containing 10% FBS or under serum deprivation (0.5% FBS) for 24 h (**A**). AGS, KATO III, and NCI-N87 cells were treated with 2.5 μg/mL tunicamycin (Tuni.) or an equal volume of DMSO for 6 h (**B**). Additionally, AGS, KATO III, and NCI-N87 cells were also transfected with HMGCS1-expressing construct (HMGCS1) or empty vector (EV) for 48 h (**B**,**C**). Whole-cell extracts of the treated cells were prepared for Western blot analysis using anti-HMGCS1, anti-p- eukaryotic translation initiation factor 2 alpha (eIF2α), anti-eIF2α, anti-activating transcription factor 4 (ATF4), anti-general control nonderepressible 2 (GCN2), anti-p-PERK, anti-PERK, and anti-GAPDH antibodies. The asterisk indicates a nonspecific band. (**D**) For indirect immunofluorescence staining, AGS cells seeded on coverslips were fixed and then stained with mouse anti-HMGCS1 and rabbit anti-PERK antibodies. HMGCS1 was detected by DyLight™ 488-conjugated goat anti-mouse IgG (green), and PERK was detected by DyLight™ 594-conjugated goat anti-rabbit IgG (red) under confocal microscopy. Cell nuclei were also visualized by staining with 4,6-diamidino-2-phenylindole dihydrochloride (DAPI) (blue). Bar, 5 μm. (**E**) Whole-cell extracts of AGS cells were immunoprecipitated with anti-HMGCS1 antibody. Then, the precipitated proteins were analyzed by Western blot analysis using anti-HMGCS1 (upper) or anti-PERK (lower) antibodies. (**F**) HMGCS1 knockout (KO) #BF4 and #FE4 cells and negative control (NC) cells were seeded for the assays of cell growth (MTT, upper), migration (middle), and invasion (lower). Cells were treated with 2.5 μg/mL tunicamycin or an equal volume of DMSO for 6 h before assessment. Mean ± SD (*n* = 3). *, *p* < 0.05; **, *p* < 0.01; ***, *p* < 0.001. ##, *p* < 0.01; ###, *p* < 0.001. (**G**) Molecular model for the nonmetabolic functions of HMGCS1 in gastric cancer progression. To enhance progression of gastric cancer cells, HMGCS1 translocates into nuclei to induce Oct4 and SOX-2 expression under stress conditions. Additionally, HMGCS1 can activate the ISR pathway to promote prosurvival signaling.

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
