# Peer review of "Mevalonate Pathway Enzyme HMGCS1 Contributes to Gastric Cancer Progression"

_cancers, 2020, doi:10.3390/cancers12051088_

Round 1
Reviewer 1 Report
The study by Wang H and collegues, “Mevalonate pathway enzyme HMGCS1 contributes to gastric cancer progression” reported that HMGCS1 promotes abilities of in vitro cell growth and progression and in vivo tumor growth and lung metastasis of gastric cancer cells.Furthermore, they reported the nuclear translocation of HMGCS1 in gastric cancer cells after serum deprivation with the binds and the activation of Oct4 and SOX-2 promoters. Moreover, the authors suggested the block of mevalonate pathway by statin and dipyridamole combination.
Has been reported that dipyridamole could suppress HMGCS1 through inhibition of cleavage of the transcription factor sterol regulatory element-binding transcription factor 2 (SREBF2). In addition, dipyridamole decreased the expression of HMGCS1, c-Myc, andcyclin D1 in colon cancer cell lines (Mol Cancer Ther; 19(1) January 2020).
Moreover, increasing evidence links ER stress and diseases associated with lipid metabolis; ER stress can lead to cholesterol accumulation via SREBP2 activation, and at the same time, increased cholesterol induce ER stress(Biochimica et BiophysicaActa 1849 (2015) 32–43).
It’s interesting the metabolic and non-metabolic role of HMGCS1 to contribute to gastric cancer progression.
The paper is well written and deals with an important issue: the rationale for a promising therapeutic strategy for the treatment of gastric cancer.
- Figure 1: the authors measured the levels of HMGCS1 in normal, tumor tissues and few lynphonodes, concern this part of the results, the authors :
- should specify the tumor staging
- should try to find, in tumor samples, the presence of HMCGCS1 in the nucleus and cytoplasm
- should specify the characteristics of the cell lines
- explain on the basis of which characteristics the authors chosen the cell lines for the experiments
- figure 1C and 1E should perform the densitometry of the wb
- Figure 3-4: the tumor volumes shown in the graphs are very small:
-explain how they were measured
-although the results seem statistically valid, four mice per group are too few to have good statistics, the authors should argue this.
-the authors should better describe the metastasis count
- Figure 5: the authors should justify the doses of the drugs used.
- Figure 6 F: the role of the stress and serum deprivation.
- authors should color cell nuclei to improve image and to better see protein shift
- The authors should test the combination of statins and dipyridamole in vivo
- The authors should discuss what patients would be chosen for treatment and above all if HMGCS1 can be used as a biomarker
- The authors should improve the references in the introduction, many are old.
Reviewer 2 Report
The article by I-Han Wang, entitled “Mevalonate pathway enzyme HMGCS1 contributes to gastric cancer progression” suggests that HMGCoA synthase exerts two independent functions, one linked to activity of the mevalonate pathway, and a second one with no direct link with this pathway, which both contribute to promote gastric cancer progression. Although the experiments are performed appropriately, and the data are rather convincing, some important questions remain that are not addressed in this study.
- It is not fair to say that statins are active against cancer, as there is a long list of arguments, including related to mis-interpretations of results from small cohorts of cancer patients, that showed the inability if statins to prevent or cure cancer alone, including appropriate prospective trials.
- Lovastatin, an HMGCoA reductase competitive inhibitor is used here at the concentration of 5 µM in gastric cancer cells, but no demonstration that the remaining level of enzyme activity is insufficient to feed the downstream lipid synthetic pathway. Even though blocking SREBP2 activity with dipyridamole should further accentuate the level of blockade, it remains that SREBP1 could still be active in controlling the activity of the mevalonate pathway. What are the effects of blocking SREBP1 on some of the endpoints surveyed? Is SREBP1 increased under conditions that suppress SREBP2? The authors should demonstrate the efficiency of blocking the mevalonate pathway under their conditions upon determining the level of mevalonate production, even in HMGCS1 KO cells.
- Along the same line, it would be important to show that the activation of Rho A, an activator of the MAPK-dependent proliferation activity is no longer prenylated to any significant extent under at least some of the different conditions (statin, dipyridamole, HMGCS1 KO of KD…).
- With respect to the hypothesis that HMGCoA synthase should possibly also localize to the nucleus, it seems that the authors made this observation by chance (and why not) as no real argument opened such a possibility. Could the authors propose strong reasons that made this hypothesis quite plausible?
- What are the minimal portions of the putative Oct4 and SOX2 promoters that bind the nuclear fraction of HMGCoA synthase? Are there sequence similarities between these regions and with other genes?
- Does HMGCoA synthase also localize to the nucleus on human cancer FFPE specimens and to what extent, in comparison with normal gastric specimens?
- Reciprocally, what is the structure of the nuclear form of HMGCoA synthase? Does it contain a nuclear localization signal? Is its nuclear translocation dependent on importin-beta? What could be the mechanism of release of HMGCoA synthase from the ER? Which segment of the protein is likely to contain a DNA-binding domain?
- Does the p53 status of the cancer lines used play a role in the nuclear activity of HMGCS synthase?
- Although the promoter CHIP assay shows binding of the protein onto DNA, it does not necessarily mean that the interaction does not require presence of an adaptor protein or nucleic acid with high affinity for the HMGCoA synthase. Demonstration of in vitro binding of purified HMGCoA synthase onto the promoter regions would presumably solve this question. Could the authors elaborate on such possibilities?
- Technically, I see no mention of the quantification method for Real-time PCR analyses.
To finish, this study is rather exhaustive and demonstrative, but it would require that these additional points be addressed appropriately to make it more convincing.
Round 2
Reviewer 2 Report
Most responses to my original concerns are satisfactory, but no experimental demonstration was provided in response to point 4. The new version of the manuscript now requires extensive English language editing.Author Response
Please see the attachment.
